# An Evidence-Based Review of Application Devices for Nitric Oxide Concentration Determination from Exhaled Air in the Diagnosis of Inflammation and Treatment Monitoring

**DOI:** 10.3390/molecules27134279

**Published:** 2022-07-03

**Authors:** Magdalena Wyszyńska, Monika Nitsze-Wierzba, Aleksandra Czelakowska, Jacek Kasperski, Joanna Żywiec, Małgorzata Skucha-Nowak

**Affiliations:** 1Department of Dental Materials, Division of Medical Sciences in Zabrze, Medical University of Silesia in Katowice, 15 Poniatowskiego Street, 40-055 Katowice, Poland; 2Department of Dental Prosthetics, Division of Medical Sciences in Zabrze, Medical University of Silesia in Katowice, 15 Poniatowskiego Street, 40-055 Katowice, Poland; mnitsze@sum.edu.pl (M.N.-W.); aczelakowska@op.pl (A.C.); protstom@sum.edu.pl (J.K.); 3Department of Clinical Pharmacology, Diabetology and Nephrology, Division of Medical Sciences in Zabrze, Medical University of Silesia in Katowice, 15 Poniatowskiego Street, 40-055 Katowice, Poland; jzywiec@sum.edu.pl; 4Department of Dental Propedeutics, Division of Medical Sciences in Zabrze, Medical University of Silesia in Katowice, 15 Poniatowskiego Street, 40-055 Katowice, Poland; mskucha-nowak@sum.edu.pl

**Keywords:** nitric oxide, inflammation process, inflammation markers, oral cavity, non-invasive diagnostics, exhaled air

## Abstract

The measurement of nitric oxide (NO) in exhaled air is used in diagnostics and monitoring the pathologies not only in the respiratory system but also in the oral cavity. It has shown a huge increase in its level in asthma and diseases of the oral cavity. It seems reasonable to undertake research on the impact of inflammation on the level of NO in exhaled air. The aim of the study is to make an evidence-based review of the application of NO levels in exhaled air in the diagnosis of inflammation and treatment monitoring on the basis of selected measuring devices. Methods and Results: This paper presents an example of the application of NO measurement in exhaled air in individual human systems. Selected measuring devices, their non-invasiveness, and their advantages are described. Discussion: The usefulness of this diagnostic method in pathologies of the oral cavity was noted. Conclusions: Measuring the level of NO in exhaled air seems to be a useful diagnostic method.

## 1. Introduction

The human body is able to function because of the systems of organs responsible for many simple or more complex processes and chemical reactions. A disturbance in the operation of at least one of them affects the entire body. One of such elements is nitric oxide (NO), which is involved in many important reactions in the human body [1].

NO is an inorganic compound in the form of a colorless gas. Initially, it was thought to be responsible mainly for environmental pollution. Knowledge in this area, based on research, has changed and many important functions have now been discovered. Reactive oxygen species (ROS) and reactive nitrogen species (RFN) can react in the cell with proteins, which in turn leads to the inactivation of enzymes and the destruction of the protein structure; with lipids (peroxidation, damage to cell membranes and organelles); with polysaccharides (disturbances in the structure of biologically active molecules); with nucleic acids (mutagenic, carcinogenic, teratogenic) (Figure 1) [2].

NO is a highly toxic, non-specific compound. In a cell, this compound is characterized by an affinity for metalloproteins and may affect the process of gene transcription in the cell nucleus, causing the activation of certain transcription factors. The exact understanding of the mechanisms of NO action in cells is extremely difficult due to the high reactivity of NO and its metabolites. NO, as a reactive form of oxygen (O_2_), interferes with the proper functioning of the entire body, including the ability to grow, repair, and respond to stress. Damaged cells and tissues stop performing their proper metabolic functions. It has been shown, however, that at moderate concentrations, RFN and ROS play an important role as mediators regulating transduction, i.e., signaling to effector cells [1,2,3,4]. NO in the human body is produced by a catalytic process with the participation of nitric oxide synthase (NOS) from L-arginine, the product of which is also L-citrulline. It is produced by three different NOS—constitutive: neuronal (nNOS-NOS I) and endothelial (eNOS-NOS III), and induced synthase (iNOS-NOS II). The NO molecule has a free radical structure and reacts with O_2_ to form NO_2_ (nitrogen dioxide) radicals—reactive forms of nitrogen (RNS). RNS obtained from the reaction of NO with a superoxide anion (O_2_•^−^) is called peroxynitrite (ONOO−), which undergoes protonation to form peroxynitric acid. This acid decomposes and forms strong oxidants: NO_2_ and O_2_•^−^. In pathological processes such as inflammation, iNOS continuously produces NO, which leads to over-activation of cyclooxygenase (COX-2) and the formation of large amounts of prostaglandins and reactive O_2_ species, which can result in excessive vasodilation and a drop in pressure. Therefore, under certain conditions, for example in septic shock, overexpression of iNOS may prove to be very detrimental to the body, and the constant exposure of cells to high concentrations of NO may have a cytotoxic effect causing damage to many organs. NO deficiency is also unfavorable and occurs in many diseases of the circulatory, gastrointestinal, urogenital, and respiratory systems [5,6,7,8,9,10].

Under physiological conditions, NO plays many important roles in the human body, including acting as a neurotransmitter in the central nervous system, as a regulator of blood flow and arterial pressure in the circulatory system, affecting the smooth muscles of the vessels—NO deficiency contributes to the development of arterial hypertension, and may also affect left ventricular hypertrophy (LVH) [5,11]. In addition, NO maintains vascular integrity through its anti-thrombotic, anti-atherosclerotic, and anti-proliferative effects. It inhibits the migration of leukocytes and cell adhesion to the endothelium and prevents the proliferation and migration of smooth muscle cells [12]. This compound plays an important role in the digestive system—it takes part in ensuring the integrity of the gastric mucosa, protecting it and regulating its blood supply, and supporting ulcer healing [13,14]. In the genitourinary system, NO regulates many aspects of normal renal homeostasis, and abnormalities in endogenous NO production can lead to nephropathy [11]. In the reproductive system, it controls steroidogenesis, folliculogenesis, and the maturation of oocytes [15]. It also takes part in sperm capacitation [16]. In the respiratory system, it dilates the bronchi, participates in the production of surfactants, and secretion of mucus, and it stimulates the mobility of the cilia [17].

Due to the multitude of functions of NO, it is also increasingly used in medicine for diagnostics. Measurement of the NO concentration in the exhaled air can be used in monitoring bronchial asthma and other general diseases. As shown in the literature, it is also a valuable diagnostic method in dentistry. However, the literature in this area is very poor in comparison to its usage in medicine (Table 1). The table presents the number of articles that include NO as a topic in the last ten years in a NO topic comparing the usage of NO in medicine, stomatology, and oral cavity pathologies. The small number of publications on the topic of NO, especially in stomatology and oral cavity pathologies, confirms the need to increase the literature resources.

The aim of the study is to make an evidence-based review of the application of NO levels in exhaled air in the diagnosis of inflammation and treatment monitoring on the basis of selected measuring devices.

## 2. Materials and Methods

### 2.1. Methods for Determining the Level of NO

NO can be detected in exhaled air (FeNO “fractional exhaled nitric oxide”, which reflects the NO level in the lower respiratory tract, from the nose nNO “nasal nitric oxide”, which reflects the level in the upper respiratory tract), as well as in saliva, urine, serum, cerebrospinal fluid or even tissue specimens. For this purpose, various methods of its detection are used: colorimetry, chemiluminescence, fluorescence, electrochemical detection, gas chromatography, electron spin resonance spectroscopy (ESR), and magnetic resonance imaging (MRI). NO concentration in saliva is measured using the Griess reaction. The Griess test is widely used for the analysis of biological samples and provides indirect detection of NO by measuring nitrites, nitrates, and nitrosating agents as an indicator for NO [18]. The following summary presents the use of portable devices for measuring NO in exhaled air. Modern devices for measuring NO in exhaled air (FeNO, nNO) are based on electrochemistry, chemiluminescence, and laser technology (Figure 2). Stationary analyzers usually measure FeNO using chemiluminescent techniques, while portable devices measure FeNO using electrochemistry, however, regardless of the measurement technology, they comply with the standardized measurement procedures recommended by the ATS (American Thoracic Society) and ERS (European Respiratory Society). In the chemiluminescence method, which is still recommended by ATS and ERS [19,20], a reaction between NO and ozone takes place inside the analyzer, and the product of this reaction is NO_2_ with an excited electron. During the transition to the ground energy state, electromagnetic radiation with a wavelength of 600–3000 nm is emitted. The emitted wave is registered by a photomultiplier, which converts it into an electrical signal. Chemiluminescence is a very sensitive measurement method—the systems used in clinical trials detect NO with a concentration of ≤1 ppb [21]. NO comes from endogenous physiochemical processes. The device has an NO filter, which eliminates NO present in the atmosphere. The flow control maintains the expiratory air flow rate at the same level regardless of the patient’s abilities. Each measurement is checked and an automatically performed test checks the isoline level. This ensures the repeatability and correctness of measurements. The so-called dead space is eliminated due to specially designed filters and the placement of the sampling port as close to the mouth as possible [22].

### 2.2. Portable Devices for Measuring NO Level in Exhaled Air

There are many portable devices on the market for testing the concentration of NO in the exhaled air. These include Aerocrine Niox, Aerocrine Niox flex, Aerocrine Niox Mino (Aerocrine, Solna, Sweden), Medisoft FeNO + (Hypair, Dinant, Belgium), as well as the CLD 88 Analyzer (Ecomedics, Durnten, Switzerland), Fenobreath (MGC Diagnostics, Saint Paul, MN, USA) and Vivatmo pro (Bosch, Waiblingen, Germany). We will describe a few of them more precisely below.

Niox Mino^®^ (Aerocrine AB, Solna, Sweden) is a portable electrochemical medical device for measuring the fraction of FeNO, providing results in parts per billion (ppb). The device meets all the requirements for NO monitoring set out by the ATS and the ERS. The device is equipped with an NO filter (scrubber), which eliminates NO present in the atmosphere. NIOX Flow Control maintains the air flow rate during exhalation at 50 mL/s regardless of the patient’s capabilities. Each measurement is checked and an automatically performed test checks the level of isolines. This ensures the repeatability and correctness of measurements. The so-called dead space is minimized in the device due to the use of unique filters and placing the sampling port as close to the mouth as possible [23].

Medisoft FeNO + (Hypair, Dinant, Belgium) is a device based on the chemiluminescent method and meets all ERS/ATS guidelines and standardized tests. The examination procedure is easy due to the intuitive software. The measurement is repeatable, with a user-replaceable built-in oxygen-free nitrogen filter. It has 3 test modes: bronchial with a flow rate of 50 mL/s, nasal with a sample flow rate of 100 mL/s, and alveolar with various flow rates (50, 100, and 150 mL/s). Values such as exhaled NO (ppb), exhaled air flow (L/min), and NO to exhaled flow (ml/min) and mean expiratory pressure (cm H_2_O) are obtained [24].

The CLD 88 analyzer (Ecomedics, Durnten, Switzerland) offers a very accurate and sensitive method for detecting FeNO and nNO. Based on chemiluminescence technology, the device delivers fast results and does not require the purchase of expensive consumables. It is approved for clinical use in Europe and fully complies with the ATS/ERS recommendations. The device measures NO flow rate and volume continuously and displays the results in real-time. Incompatible measurements are detected immediately. With the new software version, the range of applications has been extended to include numerous tests for NO analysis and general lung function assessment, suitable for infants, children, and adults [25].

Vivatmo pro (Bosch, Waiblingen, Germany) is an electrochemical device. Due to easy measurement and intuitive operation, the procedures are quick and the test results are available immediately after the measurement. The device is wireless and inductively charged. The system is maintenance-free, which means no complicated calibration procedures or costly inspections are necessary. The data can be managed directly and conveniently via the touch screen. In addition, it is integrated with healthcare IT systems due to the HL7/GDT interfaces. Bosch also has a version of the device intended for self-use by the patient (Vivatmo me) [26,27].

An important aspect of portable FeNO testing devices is the repeatability of measurements. In the study with the use of the Niox Kit, high repeatability of measurements was obtained, but it was noted that the variability of the obtained results should be at least 13% [21]. The measurements performed with the use of Medisoft, Niox, Niox Flex, and Niox Mino as well as Ecomedics devices were assessed and it turned out that not all tested analyzers gave comparable results [28]. In later studies on Vivatmo pro, Niox vero, and Ecomedics CLD 88 it was found that the measurements performed with these devices did not differ in a clinically significant way, although the measurements of Vivatmo pro were higher at higher values [20]. The subject of the research was also the stability of the results obtained with the Niox mino device. It turned out that there is a “signal drift” over time, which, although not clinically significant, indicates the need to validate each analyzer against a calibrated chemiluminescent analyzer and these procedures should be repeated at regular intervals over the life of each sensor. The measurement results range in clinically acceptable limit which is ±10 ppb according to the manufacturers of the device’s specifications. [29]. Thus, studies show that FeNO devices are not interchangeable, and most of the incompatibilities are related to higher FeNO values. Clinicians should carefully consider the application of ERS/ATS thresholds to a given device [30].

## 3. Discussion

### 3.1. Respiratory System Diseases

Measuring NO levels can be used in monitoring respiratory system diseases, digestive system diseases, and other pathologies. Over the last 15 years, the assessment of NO concentration in exhaled air has received great recognition and become very valuable in the treatment of bronchial asthma [31,32,33,34,35,36,37,38,39]. It was observed that the concentration of exhaled NO is proportional to the severity of the inflammation. In the respiratory system, the measurement of FeNO may have a wide diagnostic application. Increased concentrations are observed in asthma, allergic rhinitis, and viral infections, as well as in the rare Churg-Strauss syndrome or in the rejection of lung transplantation and obstructive sleep apnea. Reduced FeNO has been described in cystic fibrosis, primary ciliary dyskinesia, or pulmonary hypertension. Ambiguous FeNO levels are reported in bronchiectasis, COPD, and alveolar fibrosis [40,41]. Measurement of FeNO may also be useful in differentiating the etiology of cough [42,43,44]. The most important applications of FeNO measurement in respiratory diseases are described below.

#### 3.1.1. Asthma

As mentioned earlier, when tissues become inflamed, iNOS is induced, leading to the continued production of NO. In patients with asthma, an increased level of iNOS protein expression and an increased level of NO in exhaled air have been found, which is associated with eosinophilic inflammation of the respiratory tract, hence the measurement of FeNO has been found useful in monitoring the intensity of inflammation in patients with asthma [45,46]. This measurement is recommended by the ATS as part of the initial diagnosis of asthma as well as airway inflammation monitoring. Measuring FeNO can also help identify patients who are poorly controlled with their asthma, at higher risk of exacerbations, and at risk of progressive loss of lung function. Continuous patient evaluation with FeNO may be beneficial in determining the dosage of corticosteroids and monitoring patients’ compliance with corticosteroid therapy [40,46]. Moreover, this method correlates well with other indicators of the inflammatory process associated with eosinophilic infiltration, assessed in biopsy material, bronchoalveolar lavage fluid, or induced sputum, and is a simple, non-invasive and patient-friendly test [40]. The ATS guidelines define high, intermediate, and low FeNO levels in adults as >50 ppb, 25–50 ppb, and <25 ppb, respectively. In children, high, medium, and low FeNO levels are classified as >35 ppb, 20–35 ppb, and <20 ppb, respectively. According to the above guidelines, low FeNO values indicate a low probability of eosinophilic inflammation and a reaction to corticosteroids, high FeNO values indicate the probability of eosinophilic inflammation and, in symptomatic patients, a response to corticosteroids, and mean FeNO values should be interpreted with caution and in the clinical context [47,48,49,50].

#### 3.1.2. Allergic Rhinitis (AR)

AR often coexists with asthma and is also a risk factor for its development. As in asthmatics, patients with AR also have elevated FeNO levels not only during the pollen season but also outside of it [40]. A study by Antosov M et al. showed that FeNO levels did not differ between the controls and people with allergies outside the pollen season, but are significantly elevated after exposure to the allergen during the season, even if these people did not have any symptoms of the lower respiratory tract [51]. In a study involving a group of children with AR without concomitant asthma, the measured FeNO concentrations were higher compared to both the control group, patients with atopy without symptoms of rhinitis, and children suffering from non-atopic rhinitis [52]. High FeNO concentrations (>50 ppb) measured in AR patients may be a risk factor for the development of asthma in the future [42]. Increased levels of nNO during the pollen season and beyond [45] have also been shown, but it does not seem to be the optimal method of monitoring eosinophilic inflammation in the nose and sinuses, because the severity of rhinitis does not correlate with the levels of nNO detected, and these values are characterized by high variability [53]. Based on the results of research FeNO measurements are suggested to be applicable in the assessment of the severity of the inflammatory process in the lower pathways respiratory tract in patients with AR, which may allow for selecting patients who are particularly susceptible to getting asthma in the future. Additionally, they can be useful in the diagnosis of rhinitis, pointing to its allergic nature, and are associated with time duration and course of the AR [54].

#### 3.1.3. Chronic Obstructive Pulmonary Disease (COPD)

Studies on the use of NO measurement in patients with COPD have not brought unequivocal results [40,55]. They showed higher levels of FeNO in people with COPD compared to healthy controls [40,56,57]. However, previous studies did not confirm such a relationship. Measurement of FeNO in these patients can be used to predict remission, exacerbation frequency, or response to treatment, as well as to detect asthma coexisting with COPD [56]. The increase of NO concentration in patients with COPD was found in the exacerbation of the disease, which was explained by the intensification of neutrophilic bronchitis and the disturbance of the balance between oxidative factors and antioxidants [58].

#### 3.1.4. Obstructive Sleep Apnea (OSA)

The level of NO concentration was also the subject of studies by Przybyłowski et al. They investigated exhaled NO in patients suffering from OSA. The most important observation resulting from the research conducted by the authors is that the concentration of exhaled NO in patients with night apnea is much higher than in people without breathing disorders during sleep. This is probably due to the fact that people with night apnea develop inflammation of the airways [59]. OSA is recognized as an independent risk factor for cardiovascular disease and endothelial dysfunction. Hypoxaemia episodes may occur multiple times during the night and may induce oxidative stress, which causes the formation of free O2 radicals and increases the activity of pro-inflammatory factors, such as tumor necrosis factor α (TNF-α), interleukin 6 (Il-6), interleukin 8 (Il-8) or C-reactive protein (CRP) [60]. Sekosan et al. described the thickening of the lamina propria of the uvula mucosa and the existence of cellular inflammatory infiltrates in patients with obstructive sleep apnea. Rubinstein, on the other hand, showed a much higher number of polynuclear granulocytes in nasal lavage in OSA patients [61].

### 3.2. Gastrointestinal Diseases

The influence of gastrointestinal diseases on the concentration of NO also seems very interesting in the exhaled air. It is known that there are high concentrations of nitric oxide found in the sinuses and the gastrointestinal tract. The currently used devices for measuring nitric oxide in the exhaled air reduce the flow of NO from the sinuses to exhaled air. Properly functioning esophageal sphincters are supposed to protect against nitric oxide from the digestive tract [62]. The digestive tract, like the paranasal sinuses, is a reservoir in which the concentration of NO is many times higher than in the respiratory tract. As in the respiratory system, in the digestive system also FeNO measurements can be helpful in detecting inflammation of the digestive tract. The disease state detectable with this method may be located in the oral cavity and further parts of the gastrointestinal tract, although such studies are scarce. Tests were carried out with the performance of these measurements in oesophagitis, liver cirrhosis and esophageal and gastric varices, gastritis, and inflammatory bowel disease, and inconclusive data was obtained [63,64,65,66,67,68,69].

#### 3.2.1. Oral Cavity Pathologies

In our study, the presence of dental plaque significantly increased the concentration of NO in exhaled air in patients with complete dentures, and a significant correlation was also shown between the occurrence of prosthetic stomatopathy and an increased concentration of NO in the exhaled air. The presence of carious lesions, poor oral hygiene, and periodontitis also increased the concentration of NO as a marker of inflammation. In the study group, the correlation between the condition of the mucosa and the average concentration of NO in exhaled air in patients with full dentures was evaluated. The highest concentration of NO in exhaled air was observed in patients with granulomatous hypertrophic oral mucositis. Low values of NO concentration were observed in patients with no clinical symptoms of prosthetic stomatopathies. A very high level of NO concentration was observed in patients with improper denture hygiene. Patients with exemplary denture hygiene had a very low level of NO in exhaled air. The concentration of NO in exhaled air in patients with the severe carious process was on average 32 ppb, whereas in the control group including patients with no carious foci the level of the concentration was significantly lower. Among patients with exacerbated gingival bleeding, the values of GBI were high. Among patients with no gingival bleeding during the examination, the values were low. The value of the concentration of NO in exhaled air increases with the exacerbation of bleeding from the gingival pocket [66]. The study by Kamimura et al. also showed the presence of such a relationship—it suggests that oral care can reduce FeNO levels by removing bacteria and plaque and reducing the acidity of the oral environment [63]. In our own pilot study, we measured FeNO levels before and after dental treatment in a patient with profuse plaque, chronic generalized gingivitis, periodontitis, swelling, and redness of the marginal gingiva tissues near their natural teeth and implants. After comprehensive treatment, a lower FeNO level was found. This confirms that exhaled NO appears to be a valuable and easy-to-use marker of inflammation [64]. In our study, it was demonstrated that the teeth with active carious lesions turned out to have the most significant influence on the level of NO concentration in the exhaled air. This may prove that in patients affected by the caries process, numerous strains of aerobic and anaerobic bacteria are present, the presence of which increases the concentration of the inflammatory marker. Oral hygiene was also examined using the basic hygiene indicators, i.e., the plaque retention index PLI and the oral hygiene index OHI-S. Among the group of patients with improper oral hygiene, the values of NO in the exhaled air were significantly higher. Periodontium was also important for the level of exhaled NO. The gingival bleeding index according to Ainamo and Bay-GBI was used to assess the clinical condition of the periodontal tissues. In this group of patients, a significantly higher level of exhaled NO was found [66]. However, it is necessary to conduct further studies on patients with pathological conditions in the oral cavity. Currently, more studies are available, in which NO was measured in saliva, serum, or a tissue sample—in this way, it was observed, among others that the concentration of NO increased in saliva in patients with generalized chronic periodontitis [70] and in patients with oral lichen planus and recurrent aphthosis [71,72,73], as well as a NO concentration decrease in saliva with an increase in the severity of caries in children [74]. In studies in which tissue material from neoplastic lesions of the oral cavity was collected, a significant increase in tissue NO was found in squamous cell carcinoma of the oral cavity compared to the values in the control group, as well as a lower level of NO in malignant lesions compared to mild [75].

#### 3.2.2. Diseases of the Further Sections of the Digestive System

According to the literature, the level of FeNO was increased in patients with cirrhosis of the liver and esophageal and gastric varices and significantly increased in patients with cirrhosis of the liver complicated by ascites, portal vein thrombosis, red discoloration of the mucosa and portal hypertension [65]. A study of eosinophilic oesophagitis showed a slight reduction in FeNO with treatment and little significant correlation between FeNO and symptoms among responders. However, pre-treatment FeNO levels or changing FeNO levels were not found to be helpful in predicting treatment response [66]. The influence of H. Pylori infection on FeNO levels has been investigated and it has been shown that in patients with chronic gastritis infected with H. pylori the concentration of exhaled NO was increased [67]. FeNO appears to be a useful indicator of disease activity in patients with inflammatory bowel disease, especially those with ulcerative colitis [68]. In another study, however, the authors did not find that FeNO levels could be used clinically to identify individuals with inflammatory bowel disease and did not find that FeNO could be used as a marker of disease activity [69].

### 3.3. Diseases of Other Human Systems

The share of NO metabolites and NO itself has been determined in the pathogenesis of chronic sinusitis, which is confirmed by their high concentration in the exhaled air in patients suffering from this condition. The production of exhaled NO takes place in the nasal mucosa and in the paranasal sinuses [76]. Studies by other authors also confirm an increase in the level of exhaled NO as a result of tonsillitis [77]. Increased alveolar NO levels have also been demonstrated in systemic scleroderma (SSc), which is associated with the exacerbation of interstitial lung disease (ILD) [78,79]. The concentration of NO in the exhaled air was also observed in anorexia. It seems that in the case of anorexia nervosa, the levels of exhaled NO are higher, which is likely to be the result of the systemic increase in NO production due to the severe catabolic state [80].

## 4. Conclusions

NO is involved in many important processes in the human body. The FeNO test, along with other methods of measuring NO in biological samples, seems to have a great diagnostic value, especially well accepted in asthma. It is an easy, fast, non-invasive, and widely available procedure, thanks to which it can be not only a good diagnostic tool for a clinician but also a tool for monitoring disease advancement or treatment progress. However, the recommendations of the manufacturers of devices for FeNO measurements should be followed and their regular maintenance and calibration have to be taken care of to obtain precise results. More research is still needed to assess the usefulness of this measurement in medicine.

It is worth paying attention to the coexistence of pathologies related to the masticatory organ when assessing the results and their interpretation among patients visiting general medical clinics. Omission of a dental examination and the possible elimination of odontogenic foci may affect the implication of the results of general diagnostics and subsequent treatment.

## Figures and Tables

**Figure 1 molecules-27-04279-f001:**
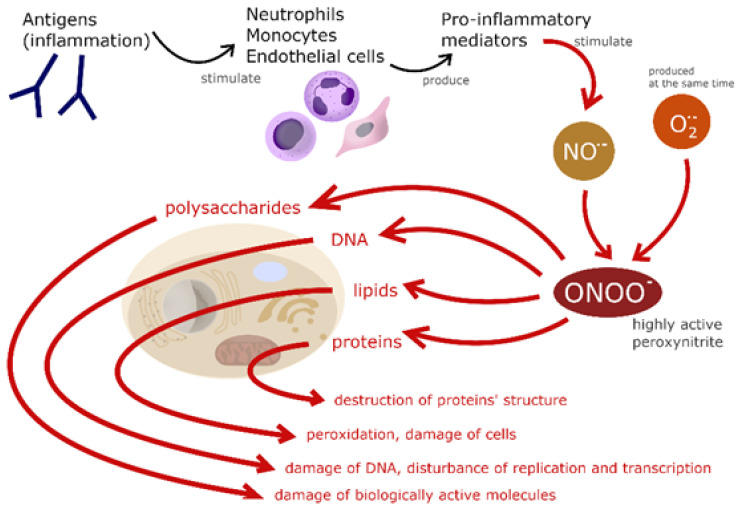
Reaction paths of ROS and RFN during oxidative stress and the negative effects of it.

**Figure 2 molecules-27-04279-f002:**
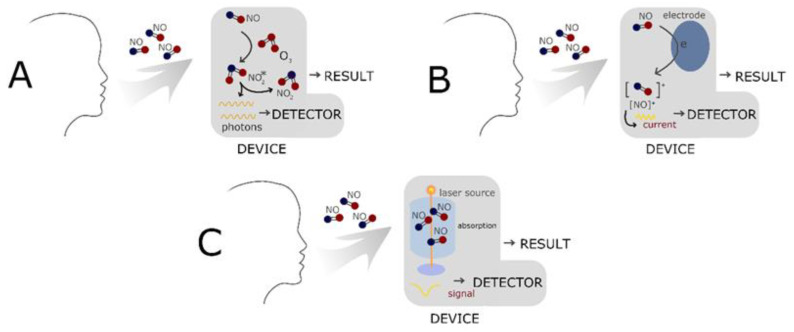
Methods of measuring NO in exhaled air: (**A**) electrochemistry, (**B**) chemiluminescence, and (**C**) laser technology.

**Table 1 molecules-27-04279-t001:** Number of literature positions in 10 years according to PubMed.

10 Years	NO in Exhaled Air in Medicine	NO in Exhaled Air in Oral Cavity Pathologies	NO in Exhaled Air in Stomatology
Review and systematic review	53	0	2
All article types	545	5	58

## Data Availability

Data supporting our results are available for request from the corresponding author.

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
