# Peer review of "An Evidence-Based Review of Application Devices for Nitric Oxide Concentration Determination from Exhaled Air in the Diagnosis of Inflammation and Treatment Monitoring"

_molecules, 2022, doi:10.3390/molecules27134279_

Round 1

Reviewer 1 Report

The authors have tried to improve the MS and [addressed] the comments raised by the reviewers. However, the MS still needs some improvements for many reasons:  

1. Introduction (P2): the authors wrote a lengthy explanation about NO, ROS and RFN in the introduction- which infact could be easily presented as a schematic reaction paths of ROS and RFN during an oxidative stress and how free radicals easily damage the cell membranes, proteins, etc. 

2. Where does NO come from? Is it endogenous or exogenous? Authors need to clearly specify the sources of NO in our body (inhaled as a pollutant gas and accumulated in cell membranes, a byproduct of many physiochemical process and/or from bacteria in our body)

3.  Materials & methods:

Fig 1 - is not clear and the fundamental principles of each method are not well presented. 

4.  Oral cavity pathologies: authors indicated that they have carried out a pilot study of FeNO level measurements in patients with profuse plaque however the authors didn't present any quantitative data or patient information. If patients were included in this experiments, did the authors received a consent from the participants? 

5. conclusions : "However, you should follow the recommendations of the manufacturers .." the use of subject pronoun, "you" is not recommended and thus the sentence need to be paraphrased. 

Reviewer 2 Report

The article, “An evidence-based review of application of determination of nitric oxide concentration in exhaled air in the diagnosis of inflammation and treatment monitoring” is a well written review article outlining the methods used to identify nitric oxide in breath samples and how increased NO concentrations can lead to different health outcomes and diseases. I recommend to publish in molecules after minor revision.

Line 33: suggest adding a keyword for “exhaled air” as it is a major focus of the article.

Lines 37-39: Sentence is convoluted and could be better written to be more concise.

Line 40: Define “NO” again at first appearance in body of paper.

Line 62: NOS has already been defined in above sentence.

Line 98: what is meant by “literature positions”? Do you mean the number of articles that include NO as a topic?

Line 102: How is stomatology specifically different than oral cavity pathologies? Stomatology is only found in Table 1 and not discussed by name within the text. Please discuss stomatology in the text if it is different and important to distinguish from oral cavity pathology.

Line 116-118: Does not seem necessary to have a section 2 on “Aim of the Study” when this is just one sentence. Could include this in the introduction; I think it has already been stated.

Figure 1: Text in figure is too small to see. Please increase size of compounds and text.

Lines 155-156: I do not think it is necessary to define ppb in scientific article.

Lines 157-158: Sentence has a strikethrough; was this supposed to be deleted?

Line 163, 167, 188: instead of using the word “thanks”, suggest using “due to”.

Line 170: be consistent with use of “NO” for nitric oxide.

Lines 174-175: Sentence has a strikethrough; was this supposed to be deleted?

Line 175: “independent of expensive consumables” sounds awkward. Do you mean “does not require purchase of expensive consumables?”

Line 183: It is unclear was “organization of the work of the office” means.

Lines 183 and 184: states twice in the same sentence that the device is “easy”. I would come up with another descriptor.

Lines 219-229: Sentence starting with “on the other hand,” does not have a verb.

Line 220: Please check spelling of “0kkkjuiCOPD”.

Lines 317-318: What data support this conclusion? Citation?

Round 2

Reviewer 1 Report

The authors have responded to the questions forwarded, the MS can be published. Few comments: 

1. Introduction ( on ROS and RFN): my previous comment still need to be considered, the fundamental reactions pathways for the formation of free radicals are necessary for any reader, recently published review by Sutaria, SR, on Metabolite 2022, 12(6) 561 can be a good reference.  

2. An opinion but not necessary is the title might need some paraphrasing:

An evidence-based review of application devices for Nitric Oxide concentration determination from exhaled air in the diagnosis of inflammation and treatment monitoring

Author Response

This manuscript is a resubmission of an earlier submission. The following is a list of the peer review reports and author responses from that submission.

Round 1

Reviewer 1 Report

The manuscript consists of a review concerning the use of exhaled NO for the diagnosis of different inflammatory pathologies and treatment monitoring.

Although the topic is interesting, the structure of the article should be reorganized and the aspects covered should be deepened.

Major concerns

ABSTRACT

In my opinion, it do not correctly summarize the aim of the paper. It is not clear if the intention of the authors is to make a general presentation on exhaled NO use in clinical application or if they want to focus on the studies concerning the feasibility of using it for the oral cavity pathologies monitoring.

INTRODUCTION

This section is clear. If possible, add a reference to the end of line 40 and use the "subscript" format for NO2 and O2

PORTABLE DEVICES FOR MEASURING NO LEVEL IN EXHALED AIR.

There is a complete lack of references to validate the information reported on the various devices.

For the Niox Mino, it is said that it is based on an electrochemical sensor and then that it is based on chemiluminescence. Please clarify.

For the Vivatmo pro, it is not reported the on which sensing method it is based.

Please clarify the concept expressed between lines 177 and 179. What is meant by the gold standard in this context? An error of 10 ppb is high if we consider that 20 ppb is the limit for considering as low the value of FeNO.

DISCUSSION

This section should be divided into subsections, based on the type of pathology addressed (e.g. respiratory, gastrointestinal, oral cavity) and the results obtained in the various studies should be presented in more detail.

Reviewer 2 Report

Wyszyńska et al present a manuscript wherein they present currently available techniques for measuring NO in exhaled air. The authors have submitted this manuscript as an article wherein one would expect either original research including own experiments and statistical analyses or a structured meta-analysis. This is not the case here.

We learn about methods and how they can be applied and there is a discussion but neither new data nor a meta-analysis of currently available studies applying these techniques. If the authors did intend to perform a meta-analysis, as they state in the aim of the study (“The aim of the study is to present the possibilities of using NO level determination in medicine in the exhaled air […] based on the data from the literature”) they should have at least provided how they searched for the studies and where in their methods section and how many studies they included and what their selection criteria were. Moreover, they would need to formulate a hypothesis and perform a statistical analysis. In this current form it cannot be considered as a meta-analysis. If they intended to submit a review than they should resubmit accordingly.

Unfortunately, I had therefore to reject this paper because it does not fit in the frame of an article though I think it is a very interesting topic. I am happy to review the manuscript again if the authors decide to resubmit.

Reviewer 3 Report

This paper lacks all sorts of scientific work!

Round 2

Reviewer 1 Report

Authors well addressed reviewer's comments. However, I would like to point out that the manucrtipt type would be "Review", as no results of a study conducted by the authors have been presented

Reviewer 3 Report

This is a literature review, device evaluation and lacks scientific work. 

Hence, I recommend the authors to expand their work and submit its as a critical review paper!